# Slow-Cycling Cells in Glioblastoma: A Specific Population in the Cellular Mosaic of Cancer Stem Cells

**DOI:** 10.3390/cancers14051126

**Published:** 2022-02-23

**Authors:** Changlin Yang, Guimei Tian, Mariana Dajac, Andria Doty, Shu Wang, Ji-Hyun Lee, Maryam Rahman, Jianping Huang, Brent A. Reynolds, Matthew R. Sarkisian, Duane Mitchell, Loic P. Deleyrolle

**Affiliations:** 1Department of Neurosurgery, University of Florida, Gainesville, FL 32611, USA; changlin.yang@neurosurgery.ufl.edu (C.Y.); guimei.tian@neurosurgery.ufl.edu (G.T.); mdajac@ufl.edu (M.D.); maryam.rahman@neurosurgery.ufl.edu (M.R.); jianping.huang@neurosurgery.ufl.edu (J.H.); brent.reynolds@neurosurgery.ufl.edu (B.A.R.); duane.mitchell@neurosurgery.ufl.edu (D.M.); 2Adam Michael Rosen Neuro-Oncology Laboratories, University of Florida, Gainesville, FL 32611, USA; 3Preston A. Wells, Jr. Center for Brain Tumor Therapy, University of Florida, Gainesville, FL 32611, USA; msarkisian@ufl.edu; 4Interdisciplinary Center for Biotechnology Research, University of Florida, Gainesville, FL 32611, USA; sun30@ufl.edu; 5Department of Biostatistics, University of Florida, Gainesville, FL 32611, USA; swang0221@ufl.edu (S.W.); jihyun.lee@ufl.edu (J.-H.L.); 6Department of Neuroscience, McKnight Brain Institute, University of Florida, Gainesville, FL 32611, USA

**Keywords:** glioblastoma, cancer stem cells, slow-cycling cells, tumor heterogeneity

## Abstract

**Simple Summary:**

A major challenge in successfully managing glioblastoma is that we do not understand the types and dynamic behaviors of the cells that constitute these tumors. Combining bioinformatics and functional studies, we describe the presence of multiple independent lineages of cancer stem cells driving the heterogeneic nature of glioblastoma. Our results help us decode and map the transcriptional and functional diversity of glioblastoma cells. By revealing potential mechanisms underlying tumor resilience, the root of resistance to treatment, our study may inform novel strategies to develop precision and effective therapies to treat brain cancer.

**Abstract:**

Glioblastoma (GBM) exhibits populations of cells that drive tumorigenesis, treatment resistance, and disease progression. Cells with such properties have been described to express specific surface and intracellular markers or exhibit specific functional states, including being slow-cycling or quiescent with the ability to generate proliferative progenies. In GBM, each of these cellular fractions was shown to harbor cardinal features of cancer stem cells (CSCs). In this study, we focus on the comparison of these cells and present evidence of great phenotypic and functional heterogeneity in brain cancer cell populations with stemness properties, especially between slow-cycling cells (SCCs) and cells phenotypically defined based on the expression of markers commonly used to enrich for CSCs. Here, we present an integrative analysis of the heterogeneity present in GBM cancer stem cell populations using a combination of approaches including flow cytometry, bulk RNA sequencing, and single cell transcriptomics completed with functional assays. We demonstrated that SCCs exhibit a diverse range of expression levels of canonical CSC markers. Importantly, the property of being slow-cycling and the expression of these markers were not mutually inclusive. We interrogated a single-cell RNA sequencing dataset and defined a group of cells as SCCs based on the highest score of a specific metabolic signature. Multiple CSC groups were determined based on the highest expression level of CD133, SOX2, PTPRZ1, ITGB8, or CD44. Each group, composed of 22 cells, showed limited cellular overlap, with SCCs representing a unique population with none of the 22 cells being included in the other groups. We also found transcriptomic distinctions between populations, which correlated with clinicopathological features of GBM. Patients with strong SCC signature score were associated with shorter survival and clustered within the mesenchymal molecular subtype. Cellular diversity amongst these populations was also demonstrated functionally, as illustrated by the heterogenous response to the chemotherapeutic agent temozolomide. In conclusion, our study supports the cancer stem cell mosaicism model, with slow-cycling cells representing critical elements harboring key features of disseminating cells.

## 1. Introduction

The cancer stem cell paradigm originated from studies of acute myeloid leukemia, which contains a subpopulation of cells showing stem-like cell properties, i.e., long-term self-renewal, the ability to generate a large number of phenotypically distinct progenies, and tumor-initiating potential [1]. Tumor cells with these features, defined as cancer stem cells, were subsequently identified in solid tumors, including GBM [2,3,4,5,6,7]. The relevance of this paradigm is supported by the functional role of CSCs in tumor growth and recurrence and the association between stem cell signature (i.e., stemness) and poor patient prognosis [8]. Diverse GBM cell populations defined phenotypically, based on the expression of markers such as CD133, CD44, ITGB8, PTPRZ1, or SOX2 [8,9,10,11,12,13,14,15,16], or based on fundamental functional characteristics, including being slow-cycling [6,7,17], exhibit the hallmark traits of CSCs described above. CD133 is a pentaspan transmembrane glycoprotein with a preferential localization to plasma membrane protrusions and microvilli, suggesting a role in membrane organization [18]. The actual function of CD133 in cancer remains elusive though. CD44, a transmembrane glycoprotein and receptor to hyaluronic acid, regulates epithelial-mesenchymal transition (EMT) and cell invasion, but its precise role in GBM is not fully understood [19]. ITGB8 is a member of the integrin beta chain complex family mediating cell–cell and cell–extracellular matrices. ITGB8 plays a role in self-renewal and survival of neural progenitor cells [20] and is associated with glioma tumor grade [21]. Shown to be required for tumorigenicity, ITGB8 regulates tumor invasion, growth, and progression by promoting TGFβ receptor signaling and mitotic checkpoint progression [15,22]. PTPRZ1 is part of the R5 subfamily of receptor-type protein tyrosine phosphatases. Shi and colleagues reported that TAM-secreted pleiotrophin activates PTPRZ1 signaling in GBM promoting CSC maintenance and tumorigenic potential [13]. PTPRZ1 knockdown attenuates the malignant properties of GBM cells and affects the expression of transcription factors such as SOX2, which is also considered a CSC marker [23]. Additionally, GBM invasive properties have been shown to be regulated by PTPRZ1 [9]. SOX2 is a transcription factor that regulates the expression of key genes and pathways involved in CSC maintenance and GBM malignancy by promoting stemness and invasiveness [12,24]. However, the precise regulatory functions of all these proteins on the maintenance and behavior of CSCs is not fully understood. Nevertheless, these markers are commonly overexpressed in cells exhibiting CSC features and have been widely used to purify such cells. 

Although the cellular fractions defined by these markers may represent overlapping populations contributing to tumorigenesis, they can also define distinct lineages of cells or cellular states with different functions regulating tumor progression and treatment resistance. Are these lineages distinct from one another? Do they exhibit a unique profile of treatment resistance? The goal of this study is to address the question of CSC population heterogeneity with a specific focus on comparing slow-cycling cancer cells with cellular populations phenotypically characterized based on the expression of defined canonical CSC markers. Our results report phenotypic, genomic, and functional profiling of these cell populations. We also demonstrate their respective clinical relevance, including comparing drug sensitivity.

## 2. Material and Methods

### 2.1. GBM Patient-Derived Cell Lines

Human GBM tumor specimens were cultured as previously described using the gliomasphere assay [6,7,17,25]. Patients consented to deposit their tissue in the brain tumor bank of the Florida Center for Brain Tumor Research of the University of Florida, from which the authors obtained de-identified specimen under a protocol approved by the Institutional Review Board committee. Cells were cultured in serum-free conditions in NeuroCult NS-A Proliferation solution with 10% proliferation supplement (STEMCELL Technologies, Vancouver, BC, Canada); Cat# 05750 and #05753) supplemented with 10 ng/mL basic fibroblast growth factor and 20 ng/mL human epidermal growth factor. 

### 2.2. Isolation of Slow-Cycling Cells

Slow-cycling cells (SCCs) were isolated as described by Hoang and colleagues [7]. Briefly, primary glioblastoma cells were labeled with CellTrace dye (Invitrogen), followed by a chase period of 5–10 days. Proliferation assessment of the cells was based on CellTrace fluorescence intensity decay rate over time measured by flow cytometry. SCCs were defined as the top 5–10% brightest cells. 

### 2.3. Antibodies

ITGB8 (R&D Systems, Minneapolis, MN, USA), MAB4775), PTPRZ1 (BD Biosciences, Franklin Lakes, NJ, USA), 610179), SOX2 (R&D Systems, MAB2018), CD44 (Biolegend, San Diego, CA, USA), 103007), and CD133 (Miltenyi Biotec, Bergisch Gladbach, Germany), 130-113-668).

### 2.4. TMZ Treatment

One day post-plating, cells were treated with 50 µM or 500 µM Temozolomide (TMZ, Sigma-Aldrich, St. Louis, MO, USA), T2577), as previously described [7]. The doses were based on the range of TMZ concentrations measured in the plasma of treated GBM patients [26,27,28] and IC50, previously reported for the line investigated in the current study [7].

### 2.5. TMZ Sensitivity Assays

After 3 days and 10 days of TMZ treatment, cocultured mCherry^+^/CD133^high^ cells and Wasabi^+^/SCCs were processed to single cells, and the ratios of Wasabi^+^/mCherry^+^ cells were measured by flow cytometry. Fixable live/dead near-infrared fluorescent reactive dye (Invitrogen, Waltham, MA, USA, L34975) was used to compare the percentages of dead cells (live/dead dye^+^) between CD133^high^ cells and SCCs in response to the different concentrations of TMZ.

### 2.6. Lentivirus Transduction

hGBM-L0 cells were transduced with vector pLV[exp]-CMV > mCherry (product ID LVS -VB191217-1841qsh-C, VectorBuilder) to constitutively express the fluorescent protein mCherry by following the manufacturer instructions. mCherry-expressing transduced cells were isolated by flow cytometry. hGBM-L0 cells virally transduced to express the Wasabi fluorescent reporter tag were kindly provided by Dr. Chang’s laboratory at the University of Florida.

### 2.7. Flow Cytometry

All flow cytometric studies were performed at the University of Florida Interdisciplinary Center for Biotechnology Flow Cytometry Core. The different cells populations were sorted out using a BD FACSAria II cytometer. BD LSR II or BD FACSymphony A3 cytometers were used for measuring and comparing cell viability and percentage of CD133^+^ ITGB8^+^, CD44^+^, PTPRZ1^+^, SOX2^+^ cells, and SCCs.

### 2.8. Bulk RNA Sequencing

RNAseq was performed as previously described [7]. SCCs isolated from nine different GBM patient-derived lines (L0, L1, L2, R24-01, R24-03, R24-23, R24-26, R24-37, and R24-47) were sequenced for paired end 150 runs. Offline data were analyzed on the University of Florida High-Performance Cluster (HiPerGator). Briefly, low-quality reads and adaptors of fastq data were trimmed by trim_galore (Babraham Bioinformatics), and then reads exceeding Q30 were aligned to Gencode v23 human genome by RSEM [29] to extract sample gene expression.

### 2.9. Single-Cell RNA Sequencing

Single-cell RNA sequencing data were derived from Darmanis and colleagues [30]. Malignant cells were selected (*n* = 1091) based on the published metadata. Genes were considered positively expressed if the mean value of TMP > 0.2. Genes expressed in less than 30 cells were excluded. The group size was determined based on the expression distribution of the different CSC markers. The number of cells per group included in the study was defined by a homogenous range of max/min ratio (MMR) of expression lacking univariate outlier using box plot methods (Appendix A). This identified the top 2% cells representing the CSC populations with the highest expression of each marker (i.e., 22 cells). Similarly, the top 2% cells for lipid metabolism and cell cycle score were used to define SCCs and FCCs, respectively. Escape package [31] was used for pathway enrichment analysis and the establishment of the lipid metabolism score. Cell cycle score was defined using the Tirosh et al. signature [7,32,33]. G1S and G2M scores were defined, and a new cell cycle score (CCS), based on the sum of G1S and G2M scores, was assigned to each cell. To visualize the level of cellular homology between groups, we used upset plots and Venn diagrams by Upset package [34] and Venny [35], respectively. Box plots for lipid metabolism signature score, CSC marker expression level, and cell cycle score were generated by ggplot2. A log10 or linear scale was applied based on the data distribution to achieve optimal visualization. Limma package [36] was used to identify differentially expressed genes (DEGs) between groups composed exclusively of private cells (*n* = 22 for SCC, *n* = 20 for FCC, *n* = 19 for SOX2, *n* = 18 for CD133, PTPRZ1, and CD44, and *n* = 17 for ITGB8). *p*-values were adjusted using Bonferroni procedure, and the significance cutoff was set at 0.005. Ggplot2 and ggrepel packages [37] were used to generate volcano plots comparing gene expression levels between SCCs and each of the other groups. The Uniform Manifold Approximation and Projection (UMAP) feature of Seurat 4.0 [38,39] was used as a deconvolution method to visualize the similarity or divergence between groups. Subsequently, a trajectory analysis was integrated with Monocle 3 package. SCCs were set as a putative start point. A gradient color scale was applied to reflect pseudotime differences. The 1000 most variable genes were identified using CancerSubtypes package [40]. A three-dimension principal component analysis was applied using Base -R. Hierarchical clustering was performed using pheatmap [41]. The DEGs between SCC and CD133 were derived as described above. A drug target enrichment between SCC and CD133 was applied by Drugbank signature (version December 2021) [42] through gene set enrichment analysis, with a cutoff of false discovery rate (FDR) set at <0.05. A heatmap (heatmap.2 package) [37] was used to visualize the predicted drug sensitivity of the groups. All codes can be obtained upon request.

### 2.10. Digital Cytometry for SCC and FCC Deconvolution

Single-cell RNAseq data of the 22 SCCs and 22 FCCs were used to construct a signature matrix by CIBERSORTx [43], which was applied for deconvolution of TCGA primary GBM bulk RNAseq data by incorporating the whole transcriptome of single cell RNAseq signature as reference. The following parameters were used in CIBERSORTx: Enable batch correction by S-mode; Disable quantile normalization; Run in absolute mode; Permutations for significance analysis: 100.

### 2.11. Hierarchical Clustering of CSC

Five CSC markers’ gene expression were subset from TCGA primary GBM dataset (TOIL). Z-scores were calculated for each CSC marker and then plotted as a heatmap with Complexheatmap [44]. Patient meta information was stacked as top annotation with SCC and FCC deconvolution scores.

### 2.12. Statistical Tests

A Wilcoxon rank sum test was applied for non-parametric pairwise comparison between reference group and each of the other groups. One-way ANOVA combined with the Bonferroni method were applied to compare cell death and cell ratio between different TMZ concentrations. *p*-values were adjusted for multiplicity using the Bonferroni method.

## 3. Results

### 3.1. Slow-Cycling Cells Express a Wide Range of CSC Markers

In GBM, we identified a subpopulation of cells displaying reduced cell cycle frequency and enriched in tumor-initiating and treatment-resistant cells exhibiting specific metabolism and enhanced infiltrative capacity [6,7,45]. Demonstration of the stemness properties in SCCs and their progenies begs the question of how this lineage compares to the population of CSCs defined based on the expression of specific markers. Multiple experimental approaches can be used to identify, isolate, and study cancer SCCs (reviewed by Basu and colleagues) [46]. We used label-retaining assays utilizing CellTrace dyes to interrogate these cells in GBM patient-derived lines [6,7,17]. In our previous study, we reported a doubling time of 73.22 +/− 7.94 h for SCCs compared to 24.96 +/− 0.87 h for the rest of the tumor cells [7]. Reduced cell division frequency of CellTrace retaining cells was also confirmed by their greater 5-ethynyl-2′-deoxyuridine (EdU) retention rate [7]. In the current study, we evaluated by flow cytometry the expression of markers commonly used to enrich CSCs such as CD44, CD133, ITGB8, PTPRZ1, and SOX2. Our results indicate that although SCCs express markers of CSCs, not 100% of them are positive, revealing some phenotypic overlap and suggesting heterogeneity and distinction between these groups of cells (Figure 1A,B). Interestingly, the expression level of these markers was similar between SCCs and the overall unselected population, except for CD133 and ITGB8, which were found to be enriched in SCCs (Figure 1B). Additionally, we isolated SCCs by FACS from nine different primary GBM patient-derived cell lines and extracted RNA to be interrogated for bulk RNA sequencing analysis. In the SCCs, we observed a wide range of expression of the different CSC markers between the nine patients (Figure 1C). Diverse degrees of CSC marker expression were also observed in FCCs and total unselected GBM cells (Appendix A). Together, these data reveal that the property of being slow-cycling and the expression of canonical CSC markers do not seem to be mutually inclusive.

### 3.2. Single-Cell Transcriptomics Identify Multiple Populations of CSCs

To further compare these populations of cells and quantify their state and potential dynamical structures, we interrogated a single-cell RNA sequencing dataset [30] and defined SCCs based on the highest score (top 2%) of a recently reported metabolic signature (Appendix A) [7], thereby identifying 22 cells (Appendix A). Similarly, CSC populations were defined by the top 22 cells with the highest expression level of CD133, SOX2, PTPRZ1, ITGB8, or CD44 (Appendix A). Fast-cycling cells (FCCs) were also included in our study and were delineated as the top 22 cells with the highest G1S/G2M cell cycle score, as previously described (Appendix A, Appendix A) [7,32,33]. We then compared the metabolic signature score, each CSC marker expression level, and cell-cycle score between all populations as defined with the criteria described above. SCCs demonstrated a significantly greater lipid metabolism signature score than every other population (Figure 2A). Interestingly, CD44^high^ cells exhibited the closest lipid score from SCCs compared to the classical CSC populations, with FCCs showing the furthest score from SCCs. Each CSC population (CD133^high^, SOX2^high^, PTPRZ1^high^, ITGB8^high^, and CD44^high^) displayed significant overexpression of their respective marker compared to the other groups (Figure 2B–F). Finally, FCCs demonstrated a higher cell cycle score than the other populations (Figure 2G). Similar to the results from our bulk RNA sequencing studies presented in Figure 1C, each of the 22 SCCs express heterogeneous expression levels of the CSC markers (CD133, SOX2, PTPRZ1, ITGB8, and CD44) ranging between several orders of magnitude (Figure 2H).

### 3.3. Heterogeneity between CSC Populations

Each group was composed of 22 cells defining populations with limited (up to 7% between CD44^high^ and ITGB8^high^ groups) to no overlap (Figure 3A and Appendix A). Upset plot indicated that SCCs represent a unique population with none of the 22 cells being included in the other groups (CD133^high^, SOX2^high^, PTPRZ1^high^, ITGB8^high^, CD44^high^, and FCC), which all share at least one cell with each other (Figure 3A and Appendix A). Twenty cells were unique to FCCs, nineteen cells were exclusive to SOX2^high^ populations, whereas eighteen cells were specific to CD133^high^, PTPRZ1^high^, and CD44^high^ cells, and seventeen to ITGB8^high^ cells. Three cells were common between PTPRZ1^high^ and CD44^high^ groups. CD133^high^ and SOX2^high^ populations shared two cells. Finally, the following paired populations had one cell in common: CD133^high^/ITGB8^high^, SOX2^high^/ITGB8^high^, PTPRZ1^high^/ITGB8^high^, ITGB8^high^/CD44^high^, CD133^high^/FCC, and ITGB8^high^/FCC. We used the dimensionality reduction technique uniform manifold approximation and projection (UMAP) [38,47] for topological comparison of the cellular fractions. We found that both SCCs and CD44^high^ cells showed tight clustering, with CD44^high^ cells being the closest neighbors of SCCs, and CD133^high^ cells being the farthest ones (Figure 3B). Even though cells were not harvested in a time series, they may be at different evolutionary positions along their lineages. We therefore performed a trajectory analysis using Monocle3 [48] to model their positions along a lineage continuum and their potential relationships with each other as a trajectory of gene expression changes (Figure 3C–E). Interestingly, our pseudotemporal cell trajectory analysis placed SCCs at one end of pseudotime (close to CD44^high^ cells) and CD133^high^ cells at the opposite, divergent end (Figure 3C). These results further support great distinctions between SCCs and CD133^high^ cells. The differential lineage positioning and pseudotime between each population were also represented as phylogenetic tree and box plot (Figure 3D,E). Gene expression was compared between SCCs and the other cell populations. We identified sets of genes differentially regulated using the limma package [36] with a cutoff of log fold change (LogFC) greater than 2 or lower than −2. (Appendix A, Appendix A). A three-dimension principal component analysis was performed with the unique cells from each group using PCA scores calculated with the top 1000 variable genes across all populations. Three-dimensional imaging plotted to visualize the linear relationship between groups indicated that CD133^high^, SOX2^high^, PTPRZ1^high^, ITGB8^high^, and FCCs were closely distributed and distant from SCCs and CD44^high^ cells, which showed greater spreading and independent clustering (Figure 3F). The expression level of these top 1000 variable genes was also represented as a heatmap, further illustrating the differential transcriptomic regulation between all of these populations (Figure 3G). Furthermore, we identified cells that we defined as CSCs based on the combined expression of two markers (Combo_CSC, Figure 2A). This population was composed of a total of nine cells simultaneously expressing high level of the following combination of CSC markers: PTPRZ1^high^/CD44^high^ (*n* = 3), CD133^high^/SOX2^high^ (*n* = 2), CD133^high^/ITGB8^high^ (*n* = 1), SOX2^high^/ITGB8^high^ (*n* = 1), PTPRZ1^high^/ITGB8^high^ (*n* = 1), and ITGB8^high^ /CD44^high^ (*n* = 1). Similar to individual CSC population, the Combo_CSC population exhibited significantly lower metabolic signature score compared to SCCs (Appendix A). Topological analysis using UMAP and evaluation of the expression level of the top 1000 variable genes also demonstrate distinct transcription profiles between SCC, Combo_CSC, and FCC populations (Appendix A). 

### 3.4. Correlation with Clinicopathological Characters and Patient Survival

We then assessed the relationship of CSC expression, SCC, and FCC characters with disease presentation. Analyzing the TCGA patient dataset, we identified that high CD44 and SOX2 expression levels were positively correlated with shorter survival (Figure 4A). In contrast, we found no difference in survival between patients characterized by high and low expression of PTPRZ1, ITGB8, and CD133 (Figure 4A). CIBERSORT was used to construct a signature matrix of SCC and FCC and for deconvolution of the TCGA primary GBM bulk RNAseq data. TCGA GBM patients were then stratified into high and low signature matrices to compare over survival. Interestingly, high SCC score was associated with poorer survival compared to a low SCC score, whereas FCC score did not predict any specific clinical outcome (Figure 4B). We further analyzed the association between the canonical CSC populations, SCCs, FCCs, and multiple clinicopathological characters of GBM by stratifying patients from the TCGA dataset based on molecular subclass combined with additional metadata including age, gender, vital status, and overall survival (Figure 4C). We found that patients with high SCC score tend to segregate in the mesenchymal molecular subtype, which showed greater SCC score associated with shorter survival (Appendix A). However, the expression level of the different CSC markers and FCC score did not correlate with additional specific clinicopathological characters. Together, these results further support intra- and inter-tumoral heterogeneity and suggest different levels of significance and relevance of the CSC lineages in disease presentation. 

### 3.5. Functional Profiling of TMZ Sensitivity

Temozolomide (TMZ) represents the standard-of-care chemotherapy used to treat GBM. The co-existence of phenotypically and functionally distinct subpopulations of cells exhibiting stemness properties may translate to a remarkable heterogeneity of drug sensitivity. To address the question of specific drug response between cell populations, we functionally compared the effect of TMZ particularly between SCCs and CD133^high^ cells. These two cell populations were isolated from a primary GBM patient line (hGBM-L0) [6,7]. mCherry-tagged CD133 cells and Wasabi-tagged SCCs were isolated by flow cytometry (Figure 5A and Appendix A) and co-cultured and treated with specific doses of TMZ (Figure 5B–E and Appendix A). Both populations of cells exhibit a different level of TMZ sensitivity, with CD133^high^ cells demonstrating significantly greater cell death compared to SCCs (Figure 5B and Appendix A), resulting in changes over time of the ratio SCC/CD133 in response to treatment (Figure 5C–E and Appendix A). The control group reflects the intrinsic proliferation rate of each population in the absence of TMZ and provides a reference for the SCC/CD133 ratio. The presence of TMZ induced significant changes in this ratio compared to controls, demonstrating different dose-dependent responses between cell populations, with greater drug tolerance by SCCs compared to CD133^high^ cells. These studies support the model of a heterogeneous pool of cells with CSC properties in GBM (i.e., SCCs vs. CD133^high^ cells), with a dynamic distribution that can be differentially regulated by therapies.

### 3.6. Genomic Profile Predicting Heterogeneity of Drug-Sensitivity

We used genomic profiling to further characterize the difference between SCCs and CD133^high^ cells and identify potential drugs predicted to target specifically SCCs versus CD133^high^ cells. DEGs between SCC and CD133^high^ groups (Figure 5F) were used for drug target enrichment identification using the comprehensive online drugbank signature database through gene set enrichment analysis (GSEA) [42]. Four drugs, including the humanized monoclonal antibody against CD44 Bivatuzumab, plasminogen activators Lanoteplase and Tenecteplase, as well as Na^+^/K^+^ ATPase inhibitor Istaroxime, were predicted to specifically target SCCs, whereas CD133^high^ cells were predicted to be sensitive to nine different other drugs (Figure 5G). These results further illustrate the intratumoral functional differences between cell lineages and encourage us to functionally investigate the effect of these drugs in future studies.

## 4. Discussion

Do populations of CSCs represent intermediate phenotypes along the spectrum of a single lineage? Do these cells present partial or complete functional redundancy with phenotypic distinction? Do CSCs exist as a homogeneous cellular population or do multiple CSCs co-exist in a given tumor? Do they reside at different stages or points of the same spectrum? Our study attempts to address these fundamental questions by further characterizing the CSC model in GBM.

Our results reveal differences in cell cycle kinetics, phenotypic and genomic profiles, and treatment sensitivity between multiple CSC populations and suggest distinct lineages or lineages with only partial overlap with a differential contribution to disease presentation and evolution. Specifically, our laboratory identified and characterized a subpopulation of slow-cycling cells in GBM [6,7,17]. These cells represent a reservoir of tumor-initiating and treatment-resistant cells exhibiting CSC properties with the ability to give rise to highly proliferative progenies maintaining lineage specificity. Even though the progenies of SCCs can exhibit similar proliferative profiles to other cancer cell populations in response to specific cues and environments, their fate seems lineage-dependent and follows distinct transcriptional trajectories [7,45]. For instance, we previously showed that freshly FAC-sorted SCCs and FCCs, which were individually intracranially implanted, gave rise to distinct progenies forming tumors with different fate characterized by specific phenotypic and metabolic profiles [7]. The present study was designed to compare SCCs with established populations of CSCs. A combination of flow cytometry and bulk and single-cell RNA sequencing analyses revealed a substantial diversity of transcriptional profiles between SCCs and cells expressing the following CSC markers CD133, CD44, ITGB8, PTPRZ1, and SOX2. Together, these results suggest that SCCs represent a distinct cell lineage with only a limited level of transcriptional redundancy.

Interestingly, we found that CD44^high^ cells were transcriptionally closer to SCCs compared to other CSC populations. This raised the question of whether selecting cells for CD44 expression can be used to purify or enrich for SCCs. The results presented in Appendix A showed a marginal increase in SCC percentages in CD44^high^ cells (11.1%) compared to CD44^low^ (7.09%), and total populations (9.31%). Based on these results, and considering that the utilization of the CellTrace retaining assay allows the collection of a pure population of SCCs (100%), CD44 expression cannot be used as a unique surrogate to define SCCs. To address the potential functional role of CD44 in SCCs, we compared the transcriptional profile of SCCs with high and low expression of CD44 (Appendix A). The absence of differentially regulated genes between both populations suggests a lack of functional differences (Appendix A). We further stratified SCC^high^ TCGA GBM patients (Figure 4) and compared the survival time between SCC^high^/CD44^high^ and SCC^high^/CD44^low^ groups. The level of CD44 expression did not influence the survival time (Appendix A). Altogether, these data strongly suggest that CD44 does not regulate SCC phenotype and function and is not sufficient for SCC selection.

To overcome the bias of selecting a single CSC marker to define a cell population and balance their potential differing importance in driving stemness, we identified a specific set of cells characterized by the concurrent high expression of a dual combination of markers. This Combo_CSC population also showed a great transcriptional difference and no cellular overlap with SCCs (Figure 3A and Appendix A). Combining multiple markers to study CSCs may allow identification of global pathways and properties of stemness; however, the advantage of studying individual CSC populations increases depth and accuracy in identifying specific targetable vulnerabilities. Of note, our study compared SCCs with only a few CSC populations; however additional markers could be selected, such as L1CAM, KLF4, integrin a6, ALDH, Nestin, Olig2, NANOG, ABCG2, or CD15 [8,10,16,49,50,51,52,53]. Importantly, due to the lack of definite, universal, and exclusive markers or functions identifying CSCs, discussion and controversy surrounding the conceptualization and contextualization of the cancer stem cell model and its hierarchical organization and regulation continue. We used a specific lipid metabolism signature, which we previously demonstrated to identify slow-cycling cells and their progenies, to classify cells as SCCs from a single cell RNA sequencing dataset [30]. We compared these cells with different CSCs populations defined based on the high expression of canonical CSC markers (i.e., CD133, ITGB8, CD44, PTPRZ1, and SOX2). Our results showed a lack of cellular overlap between SCCs and the other CSC populations, further supporting lineage specificity (Figure 3A and Appendix A).

Importantly, we established the clinical relevance of the SCC lineage. In fact, SCC signature was associated with specific clinicopathological features and demonstrated prognosis values, as indicated by poorer survival in GBM patients exhibiting strong SCC character compared to patients with low level of SCC component (Figure 4 and Appendix A).

Using a coculture dual-color system, in which CD133^+^ cells were tagged with the fluorescent protein mCherry and SCCs with the fluorescent protein Wasabi, we were able to compare in real time lineage dynamics in response to treatment. Our results indicated that SCCs and their progenies are more tolerant to TMZ than the CD133^high^ cell lineage (Figure 5 and Appendix A). Similarly, Reinartz and colleagues demonstrated specific subclone dynamism and functional consequences of intratumoral heterogeneity of drug resistance in GBM [54]. This work also supports the perspective of GBM as a disease with the co-existence of polyclonal collections of cellular hierarchies combining cancer stem cell and classical stochastic models. Oren and colleagues used a high-complexity expressed barcode lentiviral library for simultaneous tracing of cell clonal origin and proliferative and transcriptional profiling. Their results show the existence of treatment-resistant persisters in lung cancers, with their fate being lineage dependent and characterized by metabolic reprogramming of anti-oxidant and lipid pathways [55]. These data are in line with our previous study, demonstrating that, under treatment pressure, slow-cycling cancer stem cells give rise to lineage-specific cycling persisters that repopulate the tumors and are also marked by up-regulated fatty acid metabolic pathways and anti-oxidant programs [7]. These pathways, especially autophagy and lipid droplets metabolism that we reported being increased in SCCs [7], represent candidate regulators of diapause [56,57,58], which is a potential mechanism by which SCCs may enter or exhibit a drug-tolerant persister (DTP) state [59,60]. Diapause is a defined state of physiological dormancy characterized by a dormant stage of suspended embryonic development triggered by stress. Two recent studies suggest that tumor cells can engage diapause-like pathways, enabling cancer treatment escape [59,60]. Rehman and colleagues reported that colorectal cancer cells are equipotent in their ability to enter the DTP state by activating diapause-like transcriptional programs to survive therapy. Conversely, our data suggest a different scenario in GBM, which display great heterogeneity characterized by diverse populations of cells, especially cancer stem cells with distinct treatment sensitivity, suggesting a varied capacity to stimulate diapause-like mechanisms to enter the DTP state. Considering this heterogeneity, therapeutic strategies aiming to eliminate cells with stemness properties will have to be combinatorial and target every individual lineage.

Diversity in CSCs populations is now well recognized. However, the precise hierarchical organization and the plasticity of this organization between CSC populations and non-CSCs are very complex and challenging to appreciate and understand fully. One weakness of our study is the lack of depth in interrogating and modeling the dynamic aspect of the functional and phenotypic properties of tumor cells. The potential hierarchical link between SCCs and the classical CSCs could be further investigated using lineage tracing assays, similar to the report by Lan and colleagues [61]. This study used DNA barcoding and fate mapping to demonstrate a model with functionally distinct cells in GBM with a conserved proliferative hierarchy in which slow-cycling stem-like cells give rise to rapidly cycling progenitors, showing extensive self-renewing capability with the ability to generate terminally differentiated cells.

GBM are spatially organized complex ecosystems with heterogeneity across the tumor microenvironment, where specific CSC lineages may be selected based on their spatial distribution within the tumor [8], defining niche-specific cell–cell interactions. Understanding the dynamic transcriptional and spatial fluctuations of each CSC lineage and the interconnection and interconversion of these populations will be paramount for developing precision and effective therapies. The use of sophisticated high-throughput approaches, which may combine mathematical modeling, artificial intelligence, single-cell RNA sequencing, 3D model systems, multiplex imaging, and spatial transcriptomics, will help map and understand this dynamically adaptive complex system and uncover the mechanisms underlying its resilience that is the root of its resistance to treatment.

## 5. Conclusions

Our study show that cells identified as cancer stem cells based on the expression of specific markers or on particular functional criteria define distinct cell populations or states with limited overlap. Our data support a mosaicism model of cancer stem cell populations with slow-cycling cells representing critical tesserae. This multiplicity of cancer stem cell lineages further contributes to the complexity, heterogeneity, and adaptability of tumors.

## Figures and Tables

**Figure 1 cancers-14-01126-f001:**
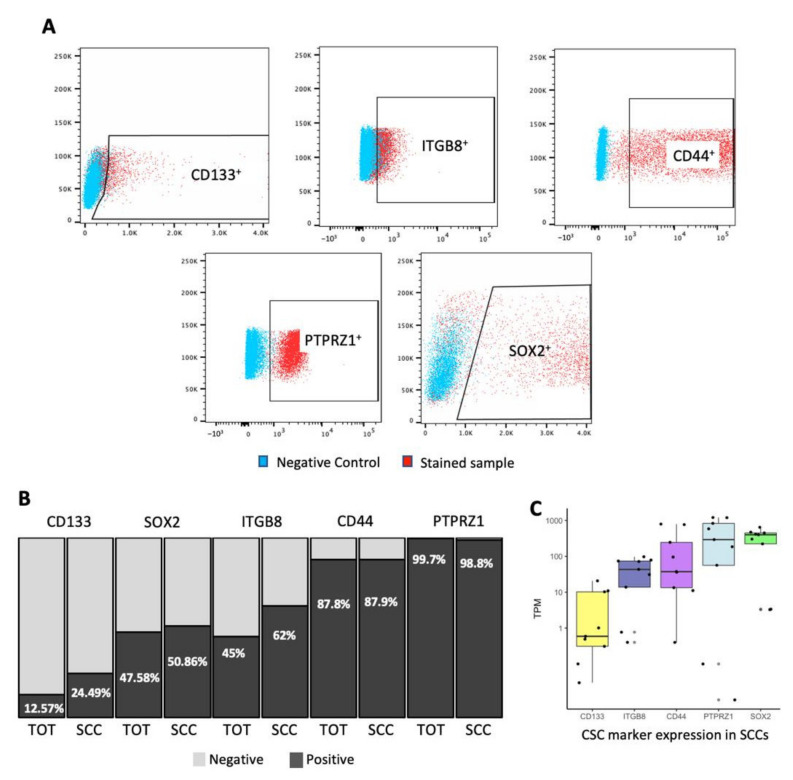
Expression level of CSCs markers in GBM SCCs. SCCs, identified as CellTrace retaining cells (top 5–10%) [6,7], were labeled with the following antibodies: anti-CD133, -ITGB8, -CD44, -PTPRZ1, and -SOX2. Protein expression was measured by flow cytometry. (**A**) Representative flow plots indicating the gates immunoreactive for the different CSC markers. (**B**) Bar graph representing the percentage of total unselected tumor cells and SCCs that are positive (dark grey) or negative (light grey) for the different CSC markers. (**C**) SCCs were FAC sorted from nine GBM patients and bulk RNA sequencing analysis was performed. The box plot indicates the level of CSC marker expression in SCCs for each patient, represented as transcript per million (TPM). The results identified SCCs in every patient and showed that SCCs exhibit a wide range of expression levels of CSC markers. Whiskers represent the 95% confidence interval and the box characterizes the interquartile range (IQR; 25th–50th–75th percentiles).

**Figure 2 cancers-14-01126-f002:**
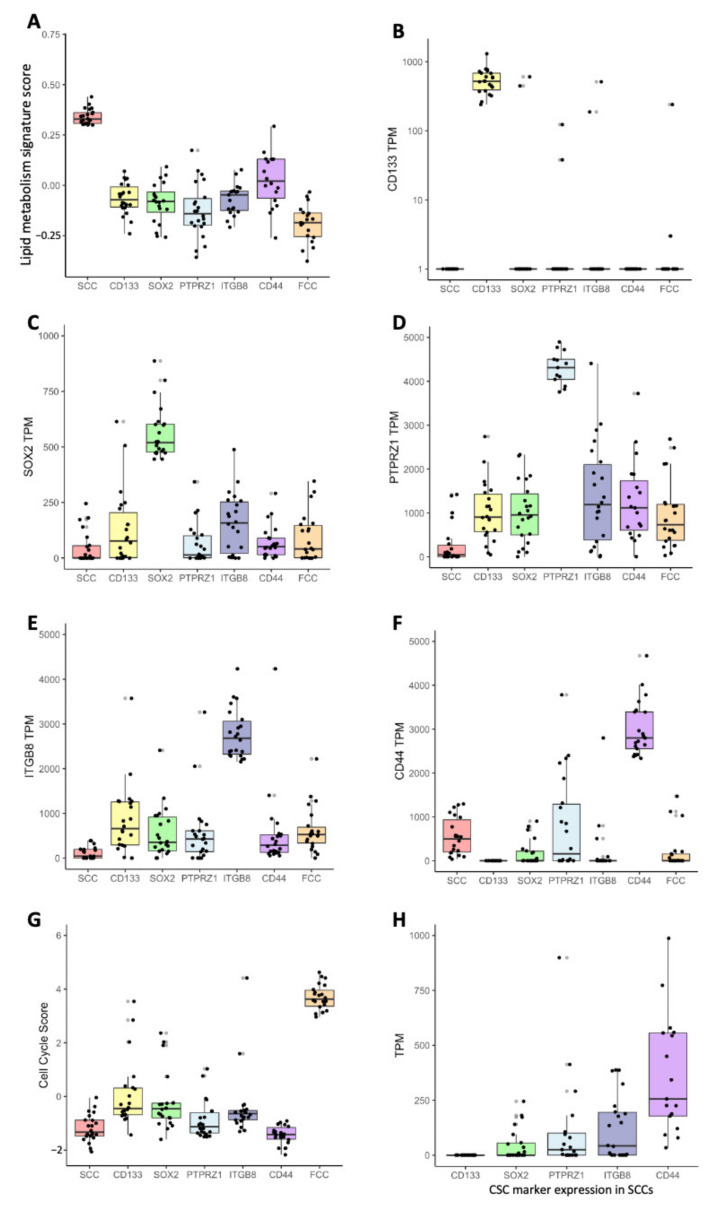
Gene signature scores and gene expression levels derived from scRNAseq comparing SCC, CSCs, and FCC groups. Deconvolution score of lipid metabolism signature (**A**), expression of CD133 (**B**), SOX2 (**C**), PTPRZ1 (**D**), ITGB8 (**E**), CD44 (**F**), and cell cycle score (**G**). All pairwise comparisons comparing groups to the reference population (i.e., SCC-(**A**), CD133-(**B**), SOX2-(**C**), PTPRZ1-(**D**), ITGB8-(**E**), CD44-(**F**), FCC-(**G**)) were statistically significant (*n* = 22, Wilcoxon test, all *p*-values adjusted for multiple comparisons using Bonferroni method were <0.001). Error bars represent the 95% confidence interval, and the box characterizes the IQR. (**H**) Expression of CSC marker in SCCs.

**Figure 3 cancers-14-01126-f003:**
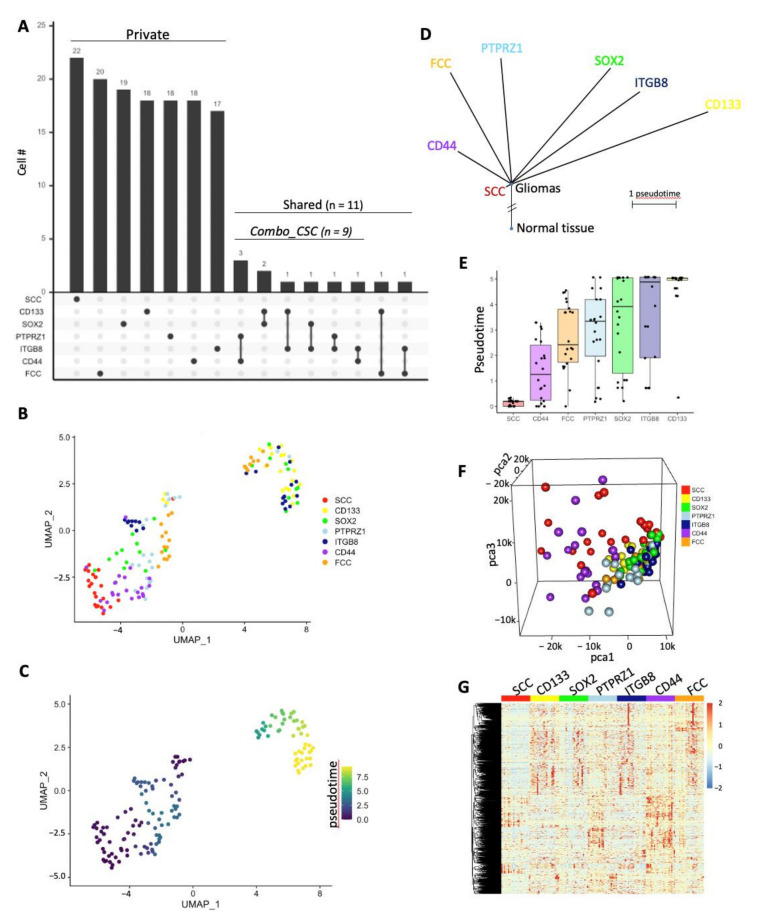
Transcriptomic differences between populations. (**A**) Upset plot showing private or shared cells among groups. Set size is 22 cells for each group. (**B**) UMAP projection of scRNA-seq data showing subsets of distinct cellular clusters. (**C**) Trajectory analysis using Monocle3 coupled with Seurat single-cell data analysis package used for UMAP projection. (**D**) Pseudotime represented using phylogenetic tree showing the evolutionary position of each lineage. (**E**) Box plot representing the pseudotime of each cell population. (**F**) Screenshot of a 3D-PCA using the top 1000 most variable genes. (**G**) Heatmap displays groups’ hierarchical clustering using the top 1000 variable genes.

**Figure 4 cancers-14-01126-f004:**
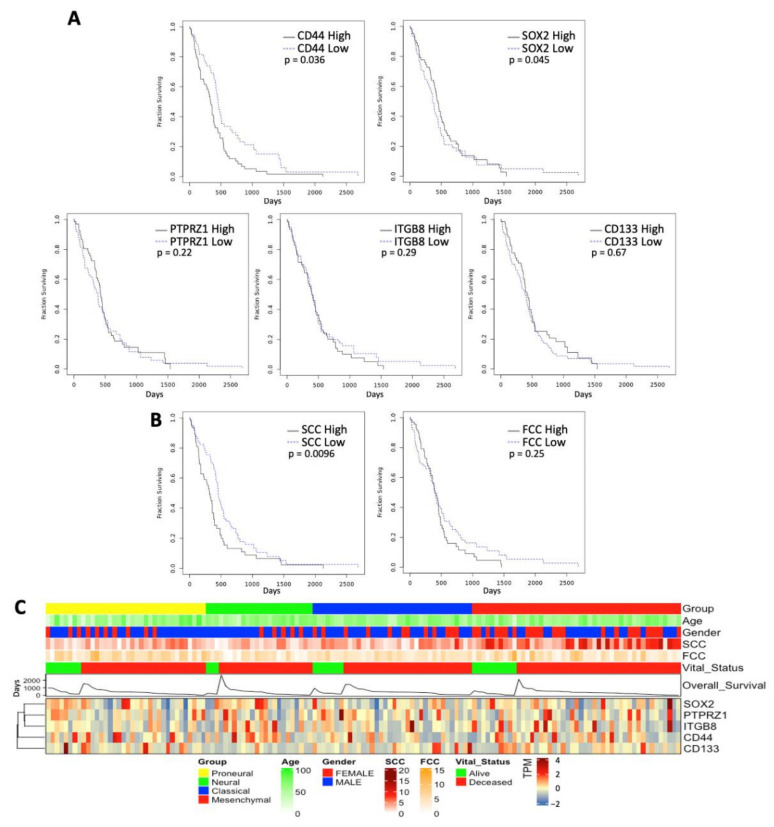
Clinicopathological characters associated with CSC lineages. Overall survival times were compared between TCGA GBM patients that were stratified by high and low expression level of the different CSC makers (mean cutoff) (**A**) and SCC and FCC deconvolution scores (**B**). (**C**) Further patient stratification was performed to discriminate between the different disease molecular subtypes (proneural, neural, classical, and mesenchymal). Additional metadata are presented (i.e., age, gender, vital status, and overall survival time).

**Figure 5 cancers-14-01126-f005:**
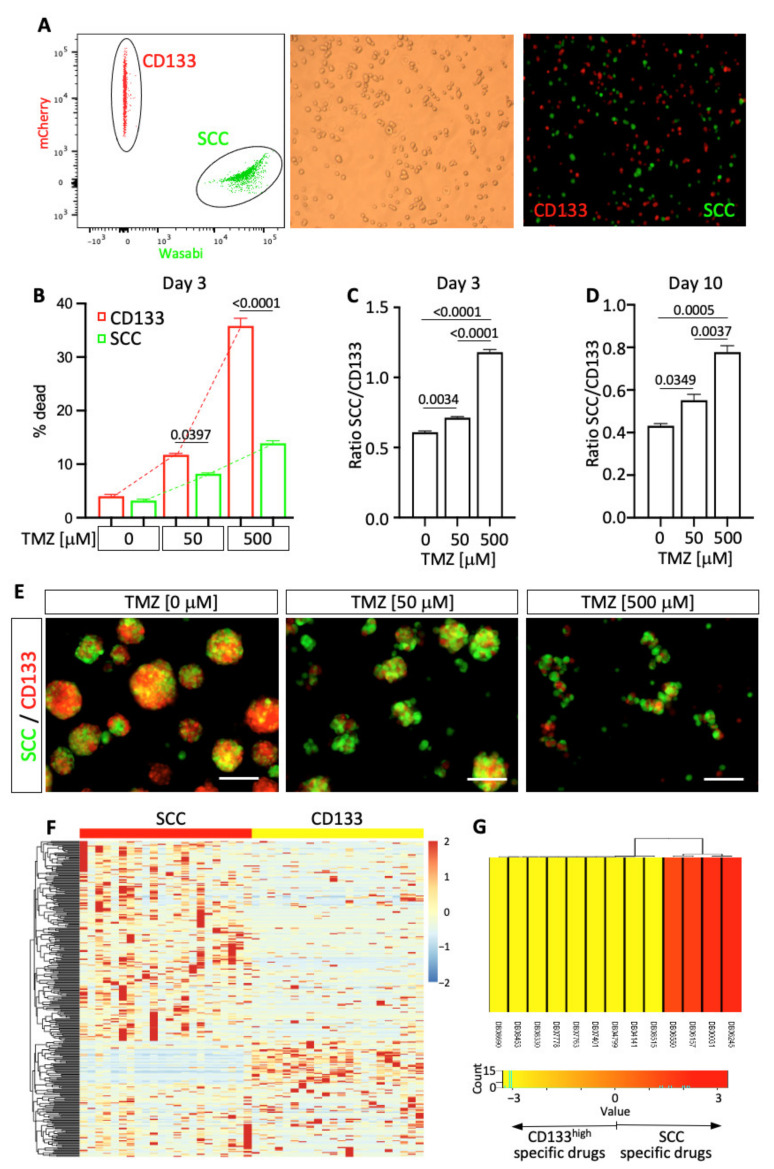
Functional assessment of the difference in drug sensitivity. (**A**) Isolated from hGBM-L0, SCCs and CD133^high^ cells were co-cultured and treated with TMZ. (**B**) Three days after initiating TMZ treatment, cell death was evaluated by flow cytometry using live/dead dye incorporation assay. Mean +/− SEM. One-way ANOVA. *p*-values were adjusted for multiplicity using the Bonferroni method. Results indicate distinct TMZ sensitivity between SCCs and CD133^high^ cells. Three days (**C**) and ten days (**D**) after TMZ treatment, the ratio SCC/CD133 was compared between the experimental conditions. Results show a significant increase in the ratio, indicating a greater resistance to TMZ of SCCs compared to CD133^high^ cells. Mean +/− SEM. One-way ANOVA. *p*-values were adjusted for multiplicity using the Bonferroni method. (**E**) Representative micrographs of co-cultured SCCs and CD133^+^ cells treated with TMZ. Scale bars, 100 µm. (**F**) Hierarchical clustering using DEGs between SCCs and CD133^high^ cells. (**G**) Drug target enrichment score between SCC and CD133^high^ groups. Drug IDs in red are agents specific to SCC. Drug IDs in yellow are specific to CD133^high^ cells. Drug ID: DB06245: Lanoteplase; DB00031: Tenecteplase; DB06157: Istaroxime; DB06550: Bivatuzumab; DB08515: (3AR,6R,6AS)-6-((S)-((S)-CYCLOHEX-2-ENYL) (HYDROXY)METHYL)-6A-METHYL-4-OXO-HEXAHYDRO-2H-FURO[3,2-C]PYRROLE-6-CARBALDEHYDE; DB04141: 2-Hexyloxy-6-Hydroxymethyl-Tetrahydro-Pyran-3,4,5-Triol; DB04799: 6-Hydroxy-5-undecyl-4,7-benzothiazoledione; DB07401: Azoxystrobin; DB07763: (5S)-3-ANILINO-5-(2,4-DIFLUOROPHENYL)-5-METHYL-1,3-OXAZOLIDINE-2,4-DIONE; DB07778: (S)-famoxadone; DB08330: METHYL (2Z)-3-METHOXY-2-{2-[(E)-2-PHENYLVINYL]PHENYL}ACRYLATE; DB08453: 2-Nonyl-4-quinolinol 1-oxide; and DB08690:Ubiquinone Q2.

## Data Availability

Data supporting the findings within this study are presented within the article and are available from the corresponding author upon reasonable request.

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
