# Peer review of "Slow-Cycling Cells in Glioblastoma: A Specific Population in the Cellular Mosaic of Cancer Stem Cells"

_cancers, 2022, doi:10.3390/cancers14051126_

Round 1

Reviewer 1 Report

The Authors aimed to characterize cell populations defined by the expression of surface markers in comparison to slow-cycling cells in order to verify the heterogeneity of stem-like cells in human GBM tumors. The paper is well written, the methods are accurately applied, and the paper is altogether sound.

A few points are worth considering.

Line 64 - what stands behind the choice of TMZ concentrations? Please, use the Greek letter for 'micro'.

Line 68 - the preparation of Wasabi expressing cells is missing. Please, add it in the methods section.

Figure 1 B and C - it would be interesting to see also the results of the expression as the percentage of all cells which are positive/negative for comparison.

Figure 2. The Authors decided the test each stem cell marker separately. While this is interesting, it would be even more interesting and informative if the Authors checked the selection of cells positive for the combination of two/three markers, e.g. CD44high/SOX2high etc.

The Authors decided to investigate five selected stem cell markers. However, these markers cannot be considered as equally important when it comes to stemness. What is the evidence of the importance of each of the markers for GBM stem cells? Also, cancer stem cells are better detected by using at least two markers during selection. This should be addressed in the Discussion.

The Authors should also describe their view of the weak points in their study.

Line 240 - the Authors decided to use co-culture in the experiment, which is quite reasonable. However, it would be interesting to compare these results with monoculture of cells. Was the ratio presented in Figure 4C affected only by sensitivity to TMZ?

Also, it would be good to the see the same type of graph for cells incubated for 10 days as for cells after 3 days incubation (Fig.4B).

Author Response

We would like to thank the reviewer for their time and effort in providing valuable comments and suggestions. Below, please find a point-by-point response to the comments:

  1. Line 64 - what stands behind the choice of TMZ concentrations? Please, use the Greek letter for 'micro'. A rational for the TMZ concentrations was provided with supporting references. The Greek letter is now provided for “micro” (Lines 96-99).
  2. Line 68 - the preparation of Wasabi expressing cells is missing. Please, add it in the methods section. hGBM-L0 virally transduced to express the Wasabi reporter tag were kindly provided by Dr. Chang’s laboratory at the University of Florida. A statement was added in the methods section (Lines 110-112).
  3. Figure 1 B and C - it would be interesting to see also the results of the expression as the percentage of all cells which are positive/negative for comparison. Figure 1B has been updated to show the percentage of the total tumor population positive/negative for the different markers measured by flow cytometry. The description of the result was updated accordingly (Lines 200-202). The expression level of the CSC markers analyzed by bulk RNA sequencing is now shown for FCCs ( Fig. 1A) and total unselected GBM cells using TCGA dataset (Supp. Fig. 1B). The text in the manuscript has been edited to describe these new data (Lines 205-207)
  4. Figure 2. The Authors decided the test each stem cell marker separately. While this is interesting, it would be even more interesting and informative if the Authors checked the selection of cells positive for the combination of two/three markers, e.g., CD44high/SOX2high etc. This is a great suggestion, and we now present such comparison in 3A and Supp. Fig. 5. We identified cells that were defined as CSCs based on the combined expression of 2 markers (Combo_CSC, Fig. 3). These cells are then compared to SCCs and FCCs (Supp. Fig. 5) (lines 283-293). A paragraph discussing this point is now included in the discussion section (Lines 404-412).
  5. The Authors decided to investigate five selected stem cell markers. However, these markers cannot be considered as equally important when it comes to stemness. What is the evidence of the importance of each of the markers for GBM stem cells? Also, cancer stem cells are better detected by using at least two markers during selection. This should be addressed in the Discussion. To overcome the bias of selecting a single CSC marker to define a cell population, and balance their respective importance in driving stemness, we identified a specific set of cells in our dataset characterized by the concurrent high expression of a dual combination of markers ( 3A, Supp. Fig. 5). This Combo_CSC population also showed a great transcriptional difference and no cellular overlap with SCCs. Combining multiple markers to study CSCs may allow identification of global pathways and properties of stemness, however, the advantage of studying individual CSC populations is the greater depth and accuracy in identifying specific targetable vulnerabilities. This paragraph was added to the discussion (Lines 404-412). Another paragraph was also included to the introduction to further develop the description of each CSC marker and their potential differential role and importance in regulating CSC behavior and GBM progression (Lines 44-65). Finally, two new figures (Fig. 4, Supp. Fig. 6) were added with the corresponding text presented in lines 302-319 to discuss the different relevance and significance of each CSC markers for disease presentation.
  6. The Authors should also describe their view of the weak points in their study. One weakness of our study is the lack of depth in interrogating and modeling the dynamic aspect of the functional and phenotypic properties of tumor cells and cancer stem cells. This is discussed in lines 460-462. A statement about the limitation and potential bias of selecting individual markers was also included in the discussion (Lines 404-412).
  7. Line 240 - the Authors decided to use co-culture in the experiment, which is quite reasonable. However, it would be interesting to compare these results with monoculture of cells. We decided to use a co-culture system in an attempt to mimic the tumor conditions where both cell populations are present. Additionally, co-cultures were chosen over monocultures to overcome the potential bias of the different cell cycle kinetic, which can influence nutrients content, for instance, and create a different environment for each population. That said, our previously published studies indicated that when cultured individually, SCCs and FCCs, for example, show different responses to TMZ (PMID: 30322894).
  8. Was the ratio presented in Figure 4C affected only by sensitivity to TMZ? We propose that the changes in cell ratio, now presented in Figure 5C-D, are dependent on the presence of TMZ since the only variable in the experiment is the concentration of the drug. The control group with no TMZ is important to note as it reflects the intrinsic proliferation rate of each population in absence of TMZ and provides a reference for the SCC/CD133 ratio. The presence of TMZ in the culture induced significant changes in this ratio compared to the no-TMZ control group, demonstrating different dose-dependent responses between cell populations. This was clarified in the result section (Lines 339-343).
  9. Also, it would be good to the see the same type of graph for cells incubated for 10 days as for cells after 3 days incubation (Fig.4B). This is a good suggestion. We have measured the cell ratio (SCC/CD133) at day 3 and day 10. These data are now shown side-by-side on the same figure as suggested ( 5C-D).

Reviewer 2 Report

The authors have investigated the heterogenous population of cells within glioblastoma tumors. Specifically they have looked at slow and fast cycling cells and compared it to cancer stem cells that express canonical stem cell markers. 

  1. The introduction needs to be expanded to include more details on cancer stem cell markers. How are these markers advantageous to cells? Why are these considered the CSC markers? The authors have a good rationale for their project but introduction to their project is lacking in detail. 
  2. How slow are slow cycling cells and how fast are fast cycling cells? BrDu assay or proliferation assay needs to be done to know doubling time of these cells. Or will authors be able to figure this out from their existing data?
  3. SCC or slow cycling cells is not introduced as an acronym. Please be sure to do that. 

Author Response

We would like to thank the reviewer for their time and effort in providing valuable comments and suggestions. Below, please find a point-by-point response to the comments:

  1. The introduction needs to be expanded to include more details on cancer stem cell markers. How are these markers advantageous to cells? Why are these considered the CSC markers? The authors have a good rationale for their project but introduction to their project is lacking in detail.The introduction was edited accordingly and now provides further information about each individual CSC marker and their reported function (Lines 44-65). Two new figures (Fig. 4 and Supp. Fig. 6) were added with descriptive text presented in lines 302-319 to illustrate the different relevance and significance of each CSC marker related to disease presentation. A paragraph in the discussion was also added in the revised manuscript to discuss these points (Lines 424-427).  
  1. How slow are slow cycling cells and how fast are fast cycling cells? BrDu assay or proliferation assay needs to be done to know doubling time of these cells. Or will authors be able to figure this out from their existing data? In our previous study, we reported a doubling time of 73.22 +/- 7.94h for SCCs compared to 24.96 +/- 0.87h for the rest of the tumor cells (PMID: 21515906). Reduced cell division frequency of CellTrace retaining cells was also confirmed by their greater 5-ethynyl-2’-deoxyuridine (EdU; similar to BrdU) retention rate (PMID: 21515906). This is now indicated in the text (Lines 192-196).
  2. SCC or slow cycling cells is not introduced as an acronym. Please be sure to do that. Thank you for pointing this out. SCCs as acronym for slow-cycling cells is now indicated in the abstract. (Line 27).

Reviewer 3 Report

Yang et al. used patient-derived GBM tumors to isolate and characterize cell populations exhibiting stemness. They use a combination of flow cytometry, bulk RNA seq, scRNA seq and phenotypic assays to good effect. In doing so, they identify that slow cycling cells are a distinct class of cells, that exhibit a distinct transcription signature, cell cycle times and drug sensitivity. While the results are useful and mostly reliable, this study has the added potential to serve as a template to inform about promising combinatorial treatments against GBM.

Some comments to address:

Lines 181-183: Explain the discrepancy between the results reported in Fig. 1C and Fig. 2H.

Lines 207-208: Explain what you mean by “developmental stages”. Are the CSCs akin to stem cells from the same tissue?

Paragraph starting Line 232:

For the experiment involving differential sensitivity of SCCs and CD133+ cells to TMZ, were the cells saturated in culture while being exposed to the drug? If so, the authors must account for or discuss the potential contribution of different cell division times to their drug sensitivity.

Discussion:

It would be useful to discuss the potential effects of implanting the slow cycling cells into a suitable host, check its tumor fate, and compare with the fast cycling or other CSCs.

Author Response

We would like to thank the reviewer for their time and effort in providing valuable comments and suggestions. Below, please find a point-by-point response to the comments:

  1. Lines 181-183: Explain the discrepancy between the results reported in Fig. 1C and Fig. 2H. The results presented in both of these figures are from different material and techniques of sequencing. Figure 1C shows gene expression level of slow-cycling cells defined functionally and isolated in vitro from patient-derived lines whereas figure 2H represents gene expression level in slow-cycling cells defined phenotypically in primary tumor tissue. Additionally, figure 1C data are from bulk RNA sequencing analysis and figure 2H data are from single cell RNA sequencing. Even though the level of gene expression and their relative distribution may differ between the figures, both results demonstrate heterogeneous levels of expression of these CSC markers in slow-cycling cells. Altogether, this supports the notion that slow-cycling properties and CSC marker expression are not mutually inclusive.
  2. Lines 207-208: Explain what you mean by “developmental stages”. Are the CSCs akin to stem cells from the same tissue? Our intent was not to refer to specific physiological developmental processes but rather to lineage stages within which cancer stem cells may be or to evolutionary positions along each lineage. Indeed, the cell stages or evolution in the lineage continuum between cell populations are likely not synchronized with each other. The trajectory analysis presented in figure 3 places cell populations with respect to each other reflecting their evolutionary position along the lineages based on gene expression changes. The term “developmental stages” was replaced by “evolutionary positions” and a more developed description is now provided (Lines 265-272). We also added new figures ( 3D-E) to further illustrate and support the different evolutionary stages of each cellular lineage based on the trajectory analysis.

Paragraph starting Line 232:

  1. For the experiment involving differential sensitivity of SCCs and CD133+ cells to TMZ, were the cells saturated in culture while being exposed to the drug? If so, the authors must account for or discuss the potential contribution of different cell division times to their drug sensitivity. To overcome the potential bias of differential saturation due to different intrinsic proliferative rate and to mimic the tumor conditions where both cell populations are present together, their respective sensitivity to TMZ was measured in a co-culture system. We propose that the changes in cell ratio, presented in Figure 4C&D, are dependent on the presence of TMZ since the only variable in the experiment is the concentration of the drug. The control group with no TMZ is important to note as it reflects the intrinsic proliferation rate of each population in absence of TMZ when cultured together and provides a reference for the SCC/CD133 ratio. The presence of TMZ in the culture induced significant changes in this ratio compared to the no-TMZ control group, demonstrating different dose-dependent responses between cell populations. This was further clarified in the result section (Lines 339-343).

Discussion:

  1. It would be useful to discuss the potential effects of implanting the slow cycling cells into a suitable host, check its tumor fate, and compare with the fast cycling or other CSCs. This is a good suggestion and represents a great experiment, which we actually recently reported (PMID: 30322894). We have edited the manuscript to describe these results in the discussion section (Lines 396-399) stating that freshly FAC sorted SCCs and FCCs, which were individually intracranially implanted, gave rise to distinct progenies forming tumors with different fates characterized by specific phenotypic and metabolic profiles.